# Brain Protease Activated Receptor 1 Pathway: A Therapeutic Target in the Superoxide Dismutase 1 (SOD1) Mouse Model of Amyotrophic Lateral Sclerosis

**DOI:** 10.3390/ijms21103419

**Published:** 2020-05-12

**Authors:** Efrat Shavit-Stein, Ihab Abu Rahal, Doron Bushi, Orna Gera, Roni Sharon, Shany G. Gofrit, Lea Pollak, Kate Mindel, Nicola Maggio, Yoel Kloog, Joab Chapman, Amir Dori

**Affiliations:** 1Department of Neurology, The Chaim Sheba Medical Center, Ramat Gan 52626202, Israel; ihab.aburahal@gmail.com (I.A.R.); doron.bushi@gmail.com (D.B.); ornagera@gmail.com (O.G.); ronisharon@gmail.com (R.S.); shanygo@gmail.com (S.G.G.); lea.pollak@gmail.com (L.P.); nicmaggio@gmail.com (N.M.); jchapman@post.tau.ac.il (J.C.); amir.dori@gmail.com (A.D.); 2Department of Neurology and Neurosurgery, Sackler Faculty of Medicine, Tel Aviv University, Tel Aviv 6997801, Israel; 3Department of Physiology and Pharmacology, Sackler Faculty of Medicine, Tel Aviv University, Tel Aviv 6997801, Israel; feingoldkate@gmail.com; 4Department of Neurobiochemistry, Weiss Faculty of Life Sciences, Tel Aviv University, Tel Aviv 6997801, Israel; kloog@post.tau.ac.il; 5Robert and Martha Harden Chair in Mental and Neurological Diseases, Sackler Faculty of Medicine, Tel Aviv University, Tel Aviv 6997801, Israel

**Keywords:** amyotrophic lateral sclerosis (ALS), superoxide dismutase 1 (SOD1), brain, thrombin, protease activated receptor 1 (PAR1)

## Abstract

Glia cells are involved in upper motor neuron degeneration in amyotrophic lateral sclerosis (ALS). Protease activated receptor 1 (PAR1) pathway is related to brain pathologies. Brain PAR1 is located on peri-synaptic astrocytes, adjacent to pyramidal motor neurons, suggesting possible involvement in ALS. Brain thrombin activity in superoxide dismutase 1 (SOD1) mice was measured using a fluorometric assay, and PAR1 levels by western blot. PAR1 was localized using immunohistochemistry staining. Treatment targeted PAR1 pathway on three levels; thrombin inhibitor TLCK (N-Tosyl-Lys-chloromethylketone), PAR1 antagonist SCH-79797 and the Ras intracellular inhibitor FTS (*S*-trans-trans-farnesylthiosalicylic acid). Mice were weighed and assessed for motor function and survival. SOD1 brain thrombin activity was increased (*p* < 0.001) particularly in the posterior frontal lobe (*p* = 0.027) and hindbrain (*p* < 0.01). PAR1 levels were decreased (*p* < 0.001, brain, spinal cord, *p* < 0.05). PAR1 and glial fibrillary acidic protein (GFAP) staining decreased in the cerebellum and cortex. SOD1 mice lost weight (≥17 weeks, *p* = 0.047), and showed shorter rotarod time (≥14 weeks, *p* < 0.01). FTS 40mg/kg significantly improved rotarod scores (*p* < 0.001). Survival improved with all treatments (*p* < 0.01 for all treatments). PAR1 antagonism was the most efficient, with a median survival improvement of 10 days (*p* < 0.0001). Our results support PAR1 pathway involvement in ALS.

## 1. Introduction

Amyotrophic lateral sclerosis (ALS) is a severe, progressive and fatal neurodegenerative disease [1,2]. A complex set of cellular events lead to the core pathology, which is motor neuron cell death. These include excitotoxicity, oxidative stress, endoplasmic reticulum stress, mitochondrial dysfunction and axonal transport disruption. Furthermore, motor neuron cell death is accompanied by inflammation [3] and interaction with a variety of glial cells, which exert either supportive or toxic effects. The heterogeneous pathogenicity of this disease is further demonstrated by both its sporadic and genetic background, where at least 25 different genes were found to be involved [4]. Due to this complex pathophysiology, devised treatment should target a common essential pathway. Indeed, Riluzole, which was proven effective in multiple pre-clinical and clinical ALS studies [5], acts by suppressing excessive motor neuron firing. Similarly, the recently introduced Edaravone acts by another general mechanism; suppressing oxidative stress. However, both treatments show limited effectiveness.

Epidemiological studies reveal that 5–10% of ALS cases are familial [6], i.e., with an inherited genetic mutation. Among the European familial ALS patients, 15% are associated with mutations in Cu/Zn superoxide dismutase 1 (SOD1) [7], which was the first gene to be identified as an etiology of genetic ALS [8]. Transgenic mice carrying mutant forms of SOD1 have been developed and serve as an accepted animal model of the disease [9,10]. Interestingly, this model demonstrates both the motor neuron dysfunction and glial involvement found in human ALS [11].

Evidence supporting the role of glia in the pathogenesis of ALS is abundant. Glial activation is seen in damaged brain regions of ALS patients [12]. SOD1 mutant astrocytes have been shown to induce damage to healthy motor neurons [13]. Knockdown of SOD1 in astrocytes derived from sporadic ALS patients prevents their toxic effects on motor neurons [14]. The neurotoxic effects found in SOD1 astrocytes include complex multi-level metabolic changes, which are not yet completely understood [15].

Activation of the coagulation pathway controls function and morphology of glia cells [16,17]. Cellular effects are mediated via the protease-activated receptor 1 (PAR1), a G-protein coupled receptor activated by serine proteases; mainly thrombin [18]. PAR1 is a junction for multiple cellular process, and its activation leads to various cellular responses [19]. As previously described, activated protein C (APC) which is one of the PAR1 agonists, mediates beneficial effects on the disease progress of SOD1 mice, via inhibition of SOD1 synthesis in motor neurons [20]. Thrombin activates PAR1 by cleavage at an alternative site resulting in biased agonism which is essentially antagonistic to the effects of APC, promoting negative effects on cell function and survival. PAR1 exerts some of its cellular effect by coupling with the Ras signaling pathway [21,22]. Under physiological conditions, we have found that PAR1 is localized to the peri-synaptic astrocytic end-feet surrounding large pyramidal neurons in the hippocampus and cerebellum of mice and rats [23]. PAR1 is involved in various CNS pathologies, including multiple sclerosis, epilepsy, neurotrauma and neoplasms [24,25,26,27]. Here we studied the astrocytic thrombin-PAR1 pathway in an animal model of ALS. Indeed, thrombin and PAR1 involvement has been described in the *wobbler* mice model for ALS [28], but not in the SOD1 mice model. We hypothesized that over-stimulation of PAR1 may be involved in astrocyte induced damage in the SOD1 mice model. To verify this hypothesis, thrombin activity along with PAR1 levels and distribution were measured in brains of SOD1 transgenic mice.

Thrombin activity was elevated in SOD1 mice brains, especially in the posterior aspects, coupled with reduced levels of PAR1. We examined the effect of thrombin-PAR1 pathway pharmacological modulation in SOD1 mice by inhibiting it at three levels: (1) inhibition of thrombin activity with *N*-Tosyl-Lys-chloromethylketone (TLCK), (2) inhibition of PAR1 activation by a direct antagonist (SCH-79797) and (3) inhibition of the PAR1 coupled Ras intracellular pathway using farnesylthiosalicylic acid (FTS), a specific signal-transduction inhibitor. These treatments improved rotarod scores to a certain degree, reduced weight loss and significantly improved survival of SOD1 mice. This suggests the PAR1 pathway as a potential intervention target for treatment of ALS.

## 2. Results

### 2.1. Increased Thrombin Activity and Decreased PAR1 Levels in Brain and Spinal Cord of SOD1 Mice

Thrombin activity was increased in all SOD1 brain slices. This increase was most pronounced in the posterior slices that correspond to the hindbrain, including the brainstem and cerebellum. These slices demonstrated a 2.5-fold higher thrombin activity in the SOD1 mice compared to healthy controls (22.7 ± 2.6 vs. 9.0 ± 1.1 mU/mL, *p* < 0.001, for slice 11). Thrombin activity was also significantly increased in slice 6 of SOD1 mice, which corresponds to posterior part of the motor cortex (12.3 ± 1.9 vs. 6.1 ± 0.8 mU/mL, respectively, *p* = 0.027). In anterior brain slices thrombin activity was non-significantly increased (10.7 ± 2.0 vs. 6.8 ± 1.3 mU/mL, *p* = 0.7 and 6.4 ± 1.4 vs. 4.3 ± 0.7 mU/mL, *p* = 0.9 for slices 3 and 4, respectively, Figure 1A). Summated thrombin activity of all brain slices was significantly higher in the SOD1 brains (94.29 ± 12.7 vs. 48.87 ± 3.7 mU/mL, *p* < 0.001, SOD1 *n* = 11, healthy control *n* = 7, Figure 1B).

Brain PAR1 protein levels measured by immunoblot were three fold lower in SOD1 mice brains compared to healthy controls (0.35 ± 0.02 vs. 1 ± 0.1, fold change, *p* < 0.01, healthy controls *n* = 5, SOD1 *n* = 4, Figure 1C). Similarly, PAR1 levels were reduced in spinal cords of SOD1 compared to healthy controls (0.59 ± 0.16 vs. 1 ± 0.1, fold change, *p* = 0.04, healthy controls *n* = 4, SOD1 *n* = 5, Figure 1D).

### 2.2. PAR1 Immunostaining in Cerebellum and Cortex

In order to further study and characterize the specific spatial decrease of PAR1 in SOD1 brains, we performed double immunofluorescence staining (SOD1 *n* = 3, healthy control *n* = 4), that was analyzed in a semi-quantitative manner. As was previously described [23] the most intense staining of PAR1 (magenta) was found surrounding the cerebellum Purkinje cells and cortex pyramidal cells (Figure 2A,I). In SOD1 mice a decrease in the intense PAR1 staining is seen in these both areas in comparison to healthy control mice (Figure 2E,M and Figure 2A,I, respectively). A co-localization is seen between astrocyte processes stained with the GFAP marker (green, Figure 2B,F,J,N), specifically surrounding the neurons (a co-localization is seen by a pink color, Figure 2C,G,K,O). These areas and structures showed reduced GFAP staining intensity in SOD1 mice, indicating loss of astrocytic processes surrounding the large neurons (Figure 2F,G,N,O). Semi-quantitative assessment of the number of Purkinje cells in the cerebellum as structures outlined by the PAR1 staining, did not show significant loss (12.0 ± 0.8 vs. 11.1 ± 0.6 cells density, *p* > 0.3). Examination of astrocytes stained for GFAP reveals large highly stained astrocytes in the granular layer in both wild type and SOD1 mice. In the SOD1 mice there are focal areas of increased GFAP staining probably constituting gliosis. In contrast, in the molecular layer of the cerebellum that ascending perpendicular pattern of astrocyte fibers staining for GFAP in the wild type animals is greatly reduced by at least 10-fold in the SOD1 mice. This staining most likely corresponds to the known structure of Bergmann glia fibers. In the present techniques it is difficult to count the number of relevant astrocytes in the molecular layer since many of these have their cell bodies in the Purkinje cell layer. Correspondingly, higher magnification of representative areas out of Figure 2C,G,K,O are presented in Figure 2D,H,L,P, respectively, indicating the intensity of PAR1 staining on the astrocyte tips was decreased in SOD1 mice both in the cerebellum (Figure 2H) and cortex (Figure 2P) in comparison to healthy control mice, indicating its loss in this disease.

### 2.3. Weight Loss, Rotarod Test and Survival

SOD1 and healthy control mice gained weight up until the age of 11 weeks (20.19 ± 0.57 and 18.88 ± 0.18 gr, respectively, *p* = 0.13, Figure 3A). Following this point, healthy control mice continued to gain weight while SOD1 mice plateaued for seven weeks, before showing progressive weight loss. At the age of 17 weeks, a significant difference in body weight was measured between healthy control and SOD1 mice (21.8 ± 0.36 and 19.99 ± 0.44 gr, respectively, *p* = 0.047). This significant weight difference persisted throughout the remaining experiment duration (*F*(50,920) = 5.93, *p* < 0.0001, SOD1 *n* = 22, healthy control *n* = 6, Figure 3A).

SOD1 mice showed a progressive decline in motor function as indicated by the rotarod test (*F*(75,1080) = 5.923, *p* < 0.0001 for the interaction, SOD1 = 21, healthy control = 5, Figure 3B). The decreased motor performances of SOD1 compared to healthy control mice began at the age of 14 weeks (42.05 ± 18.9 vs. 60 ± 0.0 s, *p* < 0.01) and became more pronounced as the experiment progressed.

SOD1 mice showed significantly reduced median survival time compared to healthy controls, as expected in this model (140 days, *p* < 0.0001, Figure 3C).

### 2.4. Treatments Effects on Body Weight, Rotarod and Survival

SOD1 mice were treated with the thrombin inhibitor TLCK, the PAR1 antagonist SCH-79797, and FTS, an inhibitor of the Ras intracellular pathway. All treatments attenuated weight loss in SOD1 mice, throughout the disease course (*F*(4,87) = 0.56, *p* = 0.69, healthy control *n* = 6, SOD1 *n* = 22, TLCK *n* = 21, PAR1 antagonist *n* = 14, FTS-40 *n* = 21, FTS-25 *n* = 14, Figure 3A). Linear regression of the effect of various treatments on the second phase of the disease showed a significant effect of both FTS-25 and PAR1 antagonist treatments (*p* = 0.026, *p* = 0.032, respectively, for the difference between the slopes, Figure 3A insert).

Treatment with FTS-40 significantly improved SOD1 mice motor performance on the rotarod at weeks 15 and 16 (56.8 ± 1.6, 31.3 ± 4.5 and 52.9 ± 31, 22.1 ± 4.4 s, *p* < 0.001 for both weeks, SOD1 mice treated with FTS-40 vs. untreated SOD1 at weeks 15 and 16, respectively, SOD1 *n* = 21, TLCK *n* = 20, PAR1 antagonist *n* = 14, FTS-40 *n* = 5, FTS-25 *n* = 13, healthy control *n* = 5, Figure 3B). Analysis of the second phase also reveals significance for PAR1 antagonist treatment compare to untreated SOD1 at week 18 (19.2 ± 6.1, 2.6 ± 1.2 s, *p* = 0.009).

Survival was significantly improved by all treatments except for high-dose FTS (FTS-60). The longest survival was achieved by treatment with the PAR1 antagonist, which increased the median survival from 140 to 150 days (*p* < 0.0001, Figure 3C). Survival was improved with TLCK, FTS-40, and FTS-25 treatments to 146, 145 and 144.5 days, respectively (*p* < 0.01, healthy control *n* = 6, SOD1 *n* = 22, PAR1 antagonist *n* = 13, TLCK *n* = 19, FTS-25 *n* = 14, FTS-40 *n* = 21, FTS-60 *n* = 19, Figure 3C).

## 3. Discussion

We report that thrombin activity is increased in brains of SOD1 mice. This is most evident in posterior structures of the brain, mainly in the brainstem and cerebellum as well as in the posterior frontal cortex, corresponding to the primary motor area. Disruption of PAR1 and GFAP astrocytic staining surrounding large pyramidal neurons is evident in some of these brain areas. High thrombin activity and glial loss of PAR1 are therefore possible treatment targets. We found that reducing the activation of the PAR1 pathway, by either blocking thrombin activity, inhibition of PAR1 or its intracellular Ras pathway improve motor performance, weight loss and survival of SOD1 mice.

It is interesting to note that the involvement of thrombin and PAR1 was already demonstrated more than 20 years ago in *wobbler* mice, another genetic model of ALS. Festoff et al. measured increased thrombin activity in cultured astrocytes derived from spinal cords of 21-day-old *wobbler* mice together with elevated PAR1 expression in their spinal cords (at P28) [28]. We assume that the decreased PAR1 levels we measured in the SOD1 mice may reflect the much later time point at which mice were sampled. This time point may represent an advanced stage of disease, by which there is an increased consumption of PAR1 by the high levels of thrombin activity. The significant changes in both models support the thrombin PAR1 pathway involvement in ALS. Involvement of the PAR1 pathway in ALS pathology is further supported by previously studies. Zhong et al. found that APC administrated systemically improves survival and reduces SOD1 transcription. This effect was found to be mediated via PAR1 and PAR3 [20]. Improved survival with APC administration, and the known inverse correlation between the effects of thrombin and APC supports the significance of our findings of elevated brain thrombin in SOD1 mice.

Our results demonstrate for the first time, elevated thrombin activity in the brain. This is supported by the previously described elevation of amyloid precursor protein (APP), a natural thrombin inhibitor, in SOD1 model [29]. The source of the increased thrombin may either be a leak through a damaged blood brain barrier (BBB) [30,31], which is a matter of debate in SOD1 model [32,33], or increased local synthesis by glial cells. The later has been described in other diseases such as in stroke [34], and by Schwann cells following a peripheral nerve injury [35]. High thrombin levels are commonly associated with low levels of PAR1 as has been found in other disorders [36,37,38]. Reduced PAR1 levels are likely due to its cleavage by thrombin and its subsequent internalization, which may lead to astrocytic and neuronal death [39,40]. Nevertheless, reduction of PAR1 levels might be secondary to astrocyte loss.

High levels of thrombin activity are neurotoxic [41], while low levels are neuroprotective, possibly through APC activation [17]. High levels of thrombin activity lowers the threshold for generating epileptic seizures in the CA3 region of the hippocampus [42], and may play a role in cognitive dysfunction [43] which is known to be affected in ALS patients. Pathological conditions such as inflammation or hemorrhage, commonly result with BBB disruption, leading to altered balance between proteases and protease inhibitors at the synapse [39]. This imbalance activates PAR1 and may attenuates astrocytic function. The main focus of the present work was the involvement of PAR1 pathway in the brain of ALS animal model. The SOD1 model demonstrates a significant involvement of the spinal cord as well, with reduction in PAR1 levels. This interesting finding calls for a future research, expanding our knowledge about the thrombin-PAR1 pathway in the spinal cord associated ALS pathophysiology.

Astrocyte are a known participant in the pathophysiology of SOD1 disease [44]. We have previously described the specific PAR1 localization on glial processes adjacent to the neuronal cell bodies and synapses. Intense PAR1 staining was especially prominent in the hippocampus and the cerebellum [23] on astrocytes around large pyramidal neurons. Here we found severe loss of PAR1 staining in SOD1 mice, specifically in astrocytes tips surrounding the pyramidal Purkinje cells. Astrocyte structure was severely disrupted as seen by the GFAP staining. Bergmann glia cells are a subtype of astrocytes that interacts with neurons in the cerebellum [45]. We suggest PAR1 staining as a possible marker for this subtype of astrocytes and their loss in the SOD1 model. Since the modifications of the cerebellum neurons and glia cells in SOD1 pathology is a highly complex issue [46], the specific PAR1 involvement is suggested for further study and more detailed characterization analysis. SOD1 mice showed reduced levels of PAR1 in their spinal cord as well, suggesting the involvement of astrocyte adjacent to motor neuron in the spinal cord.

Numerous publications describe the role of astrocytes in ALS pathophysiology. Astrocytes have an immunomodulatory effect which can be either protective or detrimental depending on disease stage [47]. SOD1 astrocytes have disrupted glutamate elimination, and glutamate toxicity may damage both adjacent astrocytes and motor neuron. Lower mitochondrial activity and changes in cytoskeleton and stellation of SOD1 astrocytes were previously described, supporting our findings of reduced GFAP staining [48]. With disease progression, further immune activation may add to the glutamate exocytotoxic effect [47]. In addition, astrocyte cross-talk is a major factor in disease pathophysiology, and is mediated by connexin CX43 expression in SOD1, which lead to enhanced gap junction coupling [49]. Here, we describe a novel mechanism in which astrocytes may contribute to ALS pathogenesis. Our results suggest detrimental over-activation of PAR1 on a specific subgroup of astrocytes associated with large pyramidal neurons. These include the upper motor neurons in the cortex, cerebellum and possibly brain stem and spinal cord. Even though the main pathological changes in ALS are classically attributed to the motor pyramidal cells of the cortex and brainstem, functional changes can be also found in other areas such as the midbrain, amygdala, hippocampus, pons and cerebellum as shown by fluorodeoxyglucose positron emission [50,51]. Cerebellar involvement in SOD1 mice model as well as in ALS patients has been previously described [52]. SOD1 mice model shows involvement of the cerebellum [53], with reduced levels of proteins such as Calbindin1 in the Purkinje cells, preceding Purkinje cell degeneration [46]. Our finding of cerebellar changes is thus not surprising.

A major question is whether thrombin-PAR1 pathway modulation play an important role in the pathogenesis of the SOD1 model. We have addressed this issue by pharmacological intervention aimed at the thrombin-PAR1 pathway at three levels: inhibition of PAR1 protease agonists, blocking the receptor itself, and blocking the major PAR1 coupled Ras-mediated downstream intracellular pathway. All three approaches improved survival, thus significantly strengthening the importance of the thrombin-PAR1 pathway.

There were some differences in the efficacy of the pharmacological approaches, with direct blockade of the receptor being the most efficient. FTS high dose (60 mg/kg) led to weight loss, as found in previous mice experiments. A detrimental effect of high FTS dosing is noted in survival studies as well. Therefore, the highest efficient dose of this drug used in mice is likely 40 mg/kg. FTS-40 treatment improved rotarod scores at the beginning of the second phase of the disease. This effect disappeared as disease progress, suggesting that the rapid general deterioration of the muscle abolishes the beneficial effect seen in motor performance. Ras pathway can be activated in several ways, including a previously described activation by endothelial growth factor (EGF), leading to reactive astrocytosis [54]. This raises the issue regarding the specificity of the beneficial effect caused by FTS. Since more specific thrombin and PAR1 up-stream pharmacological interventions resulted in improved survival as well, we hypothesize that the effect of FTS is at least partially mediated by the thrombin PAR1 pathway. Over the last decade, inhibitors of Ras have been intensively studied as promising therapeutic agents in a number of human diseases including cancer [55] and inflammatory and autoimmune diseases [56,57]. Ras inhibitors demonstrated potent anti-inflammatory properties in a number of autoimmune diseases [58,59]. Thrombin activates the Ras signaling pathway and leads to phosphorylation of CREB/ATF-1 (cAMP responsive element binding protein) and p38 [60]. pERK-immunoreactive astrocytes were detected in the cerebellum of SOD1 mice in previous reports [61], which is in line with the beneficial effects of the FTS treatment in the present study.

TLCK is a relatively non-specific thrombin inhibitor acting against a wide spectrum of proteases, rendering its effect less predictable, with a large amount of adverse effects. The dose of TLCK used in the present study was determined based on toxicity studies previously conducted in our laboratory. Doubling the dose precipitates excessive spontaneous bleeding and death in about 50% of the animals.

The dose of the PAR1 antagonist used in the present experiment is in line with the lower dosage range previous studies in mice [62]. We have found that a 100–1000 times higher single dose in normal mice is associated with seizures and behavioral dysfunction. These effects where noted with peripheral injection of PAR1 antagonist, suggesting BBB penetration. Further studies are needed to define the maximal effective concentration with an acceptable side-effect profile. It is worthwhile noting that a similar drug has been studied in humans and was found to increase the risk of intracerebral bleeding in patients with stroke [63].

This study has several limitations, including those which are inherent to the SOD1 model, and difficulties using an anti-coagulant as a therapeutic agent. ALS animal models apply only to familial forms which are less common, and the transgene must be highly over-expressed for the disease to develop, which does not necessarily represent sporadic ALS. This may simplify the involved mechanisms, or over-emphasize some of them. Most of the treatments used in this study are anti-coagulants to some degree, and therefore, may cause bleeding. However, considering the immobilization imposed upon most ALS patients in later disease stages, treating them with anticoagulants may have merits. Due to the limitation of survival studies, temporal resolution of brain thrombin activity was not measured. Although treatments improved survival, the exact association between thrombin activity, PAR1 pathway inhibition and survival requires further study. This may create a potential bias regarding the mechanism by which PAR1 pathway inhibition improves survival. Histological studies were conducted at the age of 20 weeks, and may not represent brain changes occurring later in the disease course. Levels of thrombin activity were measured only in the brain; further studies are needed to examine treatment effect on spinal cord and peripheral nervous system. Our study’s main novel finding is the involvement of the PAR1 pathway in the brain pathogenesis of ALS. The positive effect of PAR1 pathway modification only specifically supports the involvement of this pathway in the disease, but further study is needed in order to understand its role in this complex disease.

In conclusion, astrocytes are significant players in ALS pathogenesis. It remains unknown which specific reported structural, biochemical or genetic abnormality of astrocytes is the main factor in the development of the disease. To the best of our knowledge, this is the first study to demonstrate involvement of the thrombin PAR1 pathway in the pathophysiology of SOD1 mice brain. PAR1 pathway inhibition is a novel approach for treatment of ALS. Improved pharmacological approaches may be possible, including more specific inhibitors of proteases, higher doses of PAR1 antagonists or a differential treatment timeframe, combinations of drugs and direct infusion of the compounds into the brain or its surrounding fluid.

## 4. Materials and Methods

### 4.1. Animals

Mice were treated according to guidelines published by the National Institute of Health Guide for the Care and use of Laboratory Animals. This study was approved by the Sheba Medical Center animal welfare review board (permit No: 626/10/ANIM, 30/10/2010) and was conducted according to the ARRIVE (Animal Research: Reporting of In Vivo Experiments) guidelines. The study was pre-registered in the Israel Ministry of Health (MOH) registry. Mice were raised under standard conditions, 23 ± 1 °C, 12-hour light cycle (07:00 to 19:00) with access to food and water ad libitum. Humane endpoints were defined as follows: animals were sacrificed if a 10% loss of body weight was detected, or if unable to return to an upright position within 15 s after being placed on their side. Animals were assessed for health and behavior daily. If an animal has reached one of the humane endpoints described above, it was euthanized immediately using phenobarbital. Despite these measures, 33 out of the 116 animals in the survival arm of the experiment had died prior to euthanasia. In order to minimize suffering, animals were given easily accessible wet food. All research staff were qualified and certificated by the animal study committee of the MOH.

The study colony was established by crossing transgenic mice (a generous gift of Professor Groner Y., originally crossed B6SJL-Tg(SOD1*G93A)1Gur/J (the Jackson Laboratories, Bar Harbor, ME, USA) expressing high number of copies of the human homozygous *G93A* SOD1 mutation with wild-type C57BL/6 mice, for 10 generations) with wild-type C57BL/6 mice. Age of the mice in each of the study stages is presented in the Results section, as well as specific number of animals in each group. Genotyping of offspring for the SOD1 mutation was performed by PCR of DNA obtained from the tail, with the following primers: TGGGTATTGTTGGGAGGAGG, TCTGTTCCACTGAAGCTGTT. Animals that did not carry the mutation were not sacrificed, and were subjected to the control group. Animal death was monitored as part of the survival study. No Exclusion criteria were employed.

Transgenic littermates of identical genetic background served as controls and were randomly assigned to treatment groups by a blinded researcher. Both male and female mice were included in all experiments. Experiments employed adult age and gender matched transgenic male and female mice. All experiments were conducted between 8 am to 12 am and all animals were treated at the same time. No correlation was found between birth dates of the mice and life expectancy (*R*^2^ = 0.002). Motor function evaluation and fertility rate stayed constant throughout the generations, indicating stable copy number of mutated SOD1.

### 4.2. Study Design

The study included two arms. The first arm included a survival study, weight and motor evaluation. Survival was the primary outcome of this study arm. The second arm included thrombin activity, PAR1 levels and immunostaining for localization of PAR1 in the cerebellum and cortex. These were conducted at the age of 20 weeks, and were the secondary outcome of the study (Figure 1). The sample size for each treatment group is mentioned in the relevant sections of the Results. The study duration was two years.

### 4.3. Thrombin Activity

Thrombin enzymatic activity was measured using a fluorometric assay based on the cleavage rate of the synthetic flourogenic substrate Boc-Asp (OBzl)-Pro-Arg-AMC (catalog number I-1560; Bachem, Bubendorf, Switzerland) and defined by the linear slope of the fluorescence intensity versus time, as previously described [64]. Following sacrifice, brains were removed and placed in a steel brain matrix (1 mm Coronal, Stoelting, Wood Dale, IL, USA). Left and right hemispheres were separated by a midline sagittal incision. Serial coronal brain slices (1 mm thick) were prepared, beginning at slice number three, 2 mm anterior to the bregma. Each brain yielded 18 coronal slices (nine from each respective hemisphere). Each slice was placed into a single well in 96-well black microplates (Nunc, Roskilde, Denmark) containing substrate buffer (in mM: 150 NaCl, 1 CaCl_2_, 50 Tris-HCl: pH 8.0), 0.1% bovine serum albumin (BSA), bestatin 0.1 mg/mL and prolylendopeptidase inhibitor (0.2 mM). Substrate (14 µM) was added to the microplate immediately prior to initiating the fluorescence signal reading. Measurements were carried out using a microplate reader (Infinite 2000; Tecan, Männedorf, Switzerland) with excitation and emission filters of 360 ± 35 and 460 ± 35 nm, respectively. Measurements were conducted by a blinded researcher.

### 4.4. PAR1 Quantification in Mice Brains and Spinal Cords

Brain and spinal cord samples were removed as previously described [65], and were homogenized in radioimmunoprecipitation assay (RIPA) buffer (50 mM Tris HCl, pH 7.6, 150 mM NaCl, 1% NP-40, 0.5% Sodium Deoxycholate and 0.1% SDS supplied with commercial Protease Inhibitor Cocktail (P-2714, Sigma-Aldrich, Saint Louis, MO, USA) utilizing a motor homogenizer. Homogenates were incubated on ice for 10 min and centrifuged (13,000× *g,* 10 min) at 4 °C. Supernatants were collected and immediately placed on ice. Protein concentration was determined by means of a BCA (bicinchoninic acid) kit. Samples (25 µg total protein) were separated by polyacrylamide gel electrophoresis and transferred onto nitrocellulose membranes for western blot analysis. Membranes were incubated with primary rabbit anti-PAR1 antibody (1:500, abcam-ab32611 RRID:AB_778422) overnight at 4 °C, rinsed and incubated at room temperature (RT) for 1 h with horseradish peroxidase-conjugated goat anti-rabbit (1:10,000, Jackson ImmunoResearch Laboratories, West Grove, PA, USA) and bound antibody detected with enhanced chemiluminescence (ECL) assay kit (Pierce). Following detection, membranes were stripped and re-incubated at RT with mouse anti beta-actin antibody (diluted 1:10,000, 69100 MP) for 45 min followed by horseradish peroxidase-conjugated secondary antibody donkey anti-mouse (1:10,000, Jackson ImmunoResearch Laboratories, West Grove, PA, USA). Protein bands were re-detected with ECL. Bands densities were quantified with ImageJ software, a java-based image processing software (Bethesda, MD, USA) and results were normalized to actin levels. Measurements were conducted by a blinded researcher.

### 4.5. PAR1 Localization in Mice Brain Slices

To localize PAR1 in mice brains, 20 weeks old animals were anesthetized as accepted and previously described with an intraperitoneal injection of pentobarbital and perfused transcardially with phosphate buffered saline (PBS) pH 7.4, followed by 4% paraformaldehyde (PFA, Sigma-Aldrich, P6148) in 0.1 M phosphate buffer (pH 7.4). Following perfusion, brains were removed, fixed overnight in 4% PFA at 4 °C and cryoprotected by immersion in 30% sucrose in 0.1 M PO_4_ (pH 7.4) for 48 h at 4 °C. Frozen coronal sections (50 µm) were prepared with a sliding microtome (Leica, Buffalo Grove, IL, USA), collected serially and kept in a cryoprotectant buffer (24%-Glycerol, 28%-ethylene-glycol in 0.1 M PO4) at −20 °C until immunofluorescence staining. Free-floating sections were rinsed in PBST (PBS containing 0.1% Triton x-100) and blocked with 10% horse serum (diluted in PBST) for 1 h at RT. Sections were incubated with primary antibodies (rabbit anti-PAR1, Abcam ab32611, RRID:AB_778422 diluted: 1:100 and mouse anti glial-filament acidic protein (GFAP, Sigma G3893, RRID:AB_477010) diluted 1:200 in PBST containing 2% horse serum) for 48 h at 4 °C. Sections were then rinsed and incubated with the corresponding secondary antibodies (DyLight594 conjugated donkey anti-rabbit IgG and DyLight488 conjugated donkey anti-mouse IgG, Jackson immuneResearch, both diluted 1:400 in PBST containing 2% horse serum) for 1 h at RT. Sections were mounted on dry gelatin-coated slides, cover slipped with aqueous mounting medium (F4680, Sigma-Aldrich, Saint Louis, MO, USA), and viewed with a confocal microscope (Leica TCS SP5, Buffalo Grove, IL, USA) for qualitative evaluation. Quantification of Purkinje density was calculated as number of cell-bodies/200 µm field (*n* = 6 for each group). Image acquisition was conducted by a blinded technician.

### 4.6. Motor Performance

Motor performance was assessed weekly by the rotarod test, which rotated at a fixed speed of 11 rounds per minute. Mice were permitted to run for up to 1 min on each trial, or until they fell off. A mean of three consecutive trials was calculated for each mouse. Motor performance, weekly weighing and survival follow-up were all conducted by a blinded researcher.

### 4.7. Pharmacological Interventions

Treatments were administrated five times a week. FTS (*S*-trans-trans-farnesylthiosalicylic acid) was administrated orally (volume of 100 µL) in low and high dosages (25, 40 and 60 mg/kg, respectively) utilizing an appropriate gavage. TLCK (*N*-Tosyl-Lys-chloromethylketone) was used at a dose of 4.4 mg/kg and PAR1 antagonist (SCH-79797, Tocris, Bristol, UK) was used at a dose of 25 μg/kg, both were administered by intraperitoneal injections (Figure 4).

Animal sample size used in the study: thrombin activity: SOD1 *n* = 11, healthy control *n* = 7. PAR1 levels: healthy control *n* = 5, SOD1 *n* = 4. Immunostaining: SOD1 *n* = 3, healthy control *n* = 4. Weight: healthy control *n* = 6, SOD1 *n* = 22, TLCK *n* = 21, PAR1 antagonist *n* = 14, FTS-40 *n* = 21, FTS-25 *n* = 14. Rotarod: SOD1 *n* = 21, TLCK *n* = 20, PAR1 antagonist *n* = 14, FTS-40 *n* = 5, FTS-25 *n* = 13, Healthy control *n* = 5. Survival: SOD1 *n* = 22, PAR1 antagonist *n* = 13, TLCK *n* = 19, FTS-25 *n* = 14, FTS-40 *n* = 21, FTS-60 *n* = 19, healthy control *n* = 6.

### 4.8. Statistics

Sample size was based on previous studies [66]. Changes in thrombin activity were analyzed by Mann–Whitney test. Changes in PAR1 levels were calculated using *t*-test, normality was evaluated using the Kolmogorov–Smirnov test.

The SOD1 disease includes two phases. The first phase, lasting up until week 16, includes a relative stable weight, and the second phase includes a fast deterioration in general health and weight. Therefore, analyses of the clinical parameters (weight, rotarod score) were divided to weeks 5–15 and 16–20, named first and second disease phases, respectively. Since each treatment affect different component of the PAR1 pathway, analyses of clinical parameters and survival analysis included evaluation of all treatments combined, and of each treatment separately.

Changes in the rotarod test were analyzed using two-way ANOVA with Tukey post-hoc analysis, for the entire disease course and separately for the second phase. Changes in body weight were analyzed by two-way ANOVA with Tukey post-hoc analysis, and by individual linear regression of each treatment for the second phase of the disease (weeks 16–20). Normality was evaluated using the Kolmogorov–Smirnov test.

Survival analysis was calculated using log-rank (Mantel–Cox) test with Bonferroni method for multiple comparisons correction. All calculations were made using GraphPad Prism (version 8.0 for Windows, GraphPad Software, La Jolla, CA, USA, www.graphpad.com).

## Figures and Tables

**Figure 1 ijms-21-03419-f001:**
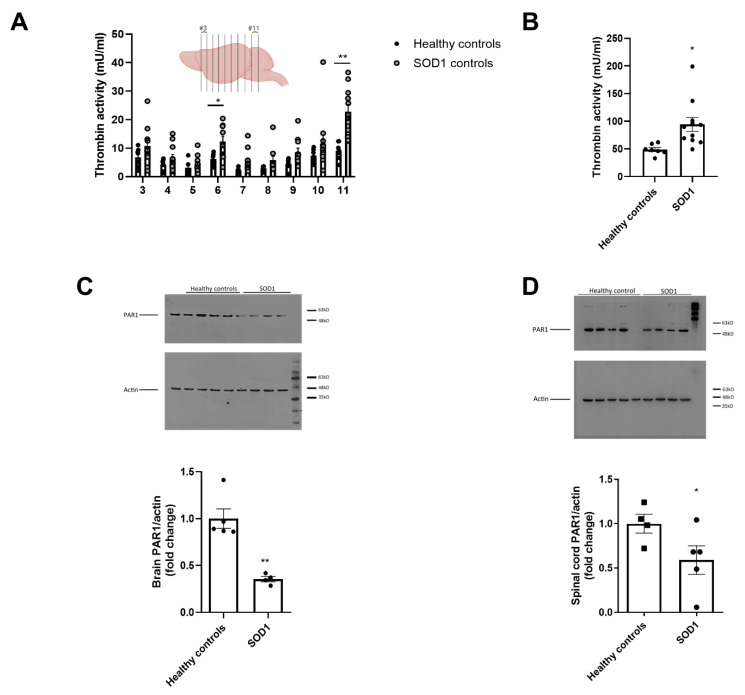
Thrombin activity and protease activated receptor 1 (PAR1) levels in superoxide dismutase 1 (SOD1 mice): (**A**) thrombin activity is significantly increased in the posterior brain slices, as well as posterior frontal cortex (slice 11, 6, *p* < 0.001, *p* < 0.05, respectively). Measured from the anterior slice number 3 (#3) to the posterior slice number 11 (#11). Number of animals: SOD1 *n* = 11, healthy control *n* = 7; (**B**) summation of thrombin activity in all brain slices of SOD1 mice is significantly elevated compared to healthy control mice (*p* = 0.013). Number of animals: SOD1 *n* = 11, healthy control *n* = 7; (**C**) PAR1 levels, as measured by western blot, are significantly lower in SOD1 mice brains compared to healthy control mice (*p* < 0.01). Number of animals: SOD1 *n* = 4, healthy control *n* = 5; (**D**) PAR1 levels in spinal cords of SOD1 mice were lower compared to healthy control mice. Number of animals: SOD1 *n* = 5, healthy control *n* = 4. * *p* < 0.05, ** *p* < 0.01. Illustration created using BioRender.com.

**Figure 2 ijms-21-03419-f002:**
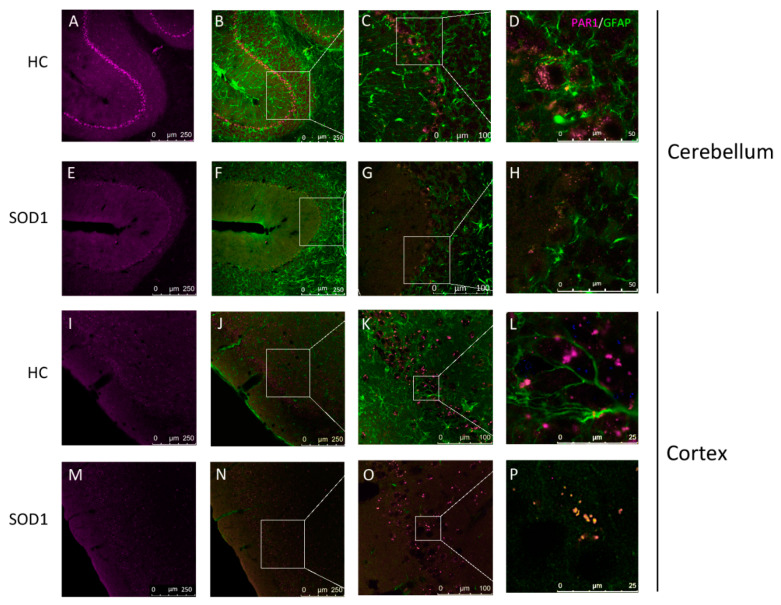
PAR1 localization in mice cerebellum and cortex: Confocal fluorescent microscopy immunohistochemistry with antibodies labeling PAR1 (magenta) and the astrocytic marker glial acidic fibrillary protein (GFAP, green) in (**A**–**H**) the cerebellum and (**I**–**P**) the cortex of healthy and SOD1 mice in ×20 magnification. Co-localization of PAR1 with GFAP is seen in pink. Further magnification of ×63 is shown in C,G,K,O. High power magnification of selected areas (indicated by white frames) are shown in D,H,L,P. SOD1 mice show reduced levels of PAR1, and disruption of its co-localization with GFAP, as was indicated by qualitative analysis. Reduction of GFAP staining suggests astrocytic morphological modification. Immunohistochemistry was performed at the age of 20 weeks. All photos had color adaptation using Adobe Photoshop CS5. Number of animals: SOD1 *n* = 3, healthy control *n* = 4.

**Figure 3 ijms-21-03419-f003:**
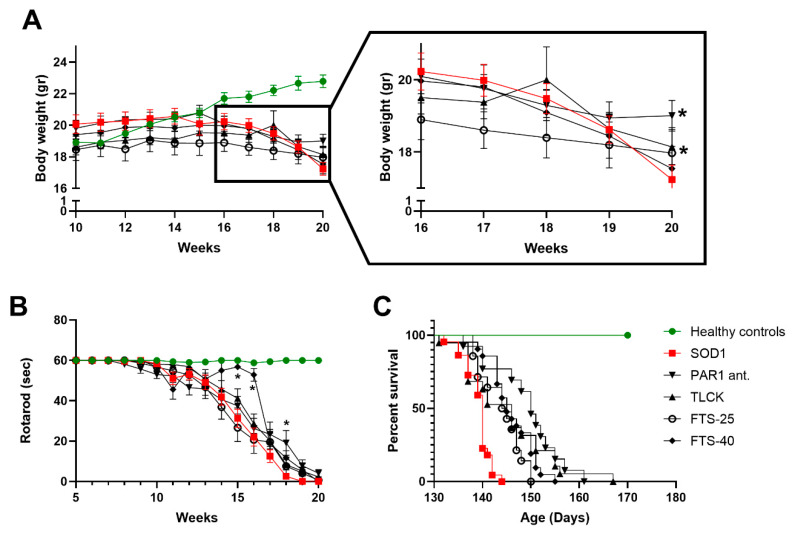
Body weight, rotarod and survival outcomes of SOD1 mice with various treatment: (**A**) healthy control mice progressively gain weight, while SOD1 mice gain weight up until the age of 11 weeks. Following a plateau, at the age of 17 weeks, SOD1 mice show progressive weight loss. A significant difference in body weight is measured between healthy control and SOD1 mice. This difference persists throughout the remaining experiment duration (*p* < 0.0001). In the insert, an enlargement of weeks 16–20 is presented. All treatments show a beneficial effect on weight loss. Number of animals: healthy control *n* = 6, SOD1 *n* = 22, *N*-Tosyl-Lys-chloromethylketone (TLCK) *n* = 21, PAR1 antagonist *n* = 14, S-trans-trans-farnesylthiosalicylic acid (FTS-40) *n* = 21, FTS-25 *n* = 14; (**B**) SOD1 mice show shorter rotarod scores starting at the age of 14 weeks and subsequently thorough the experiment (*p* < 0.0001). FTS-40 improves rotarod scores at weeks 15 and 16 (*p* < 0.01). Number of animals: Healthy control *n* = 5, SOD1 *n* = 21, TLCK *n* = 20, PAR1 antagonist *n* = 14, FTS-40 *n* = 5, FTS-25 *n* = 13; (**C**) Survival is significantly improved by all treatments except for high-dose FTS (*p* < 0.01). Number of animals: Healthy control *n* = 6, SOD1 *n* = 22, PAR1 antagonist *n* = 13, TLCK *n* = 19, FTS-25 *n* = 14, FTS-40 *n* = 21, FTS-60 *n* = 19. * *p* < 0.05.

**Figure 4 ijms-21-03419-f004:**
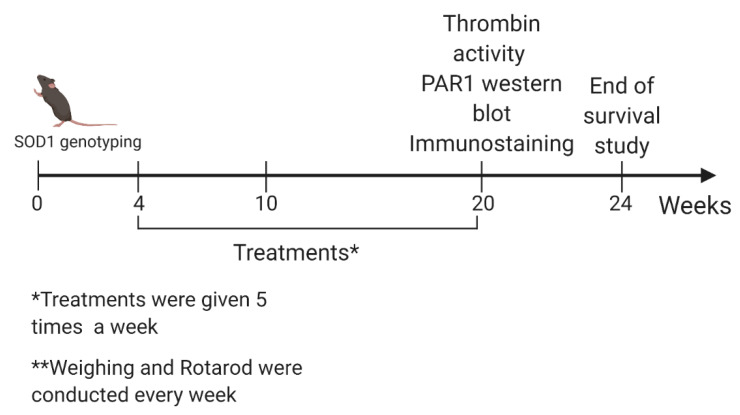
Timeline: transgenic mice were genotyped at birth. Weighing and rotarod evaluation were performed every week. Treatments were given at weeks 4–20, five times a week. Thrombin activity, western blot for PAR1 and immunostaining were conducted at the age of 20 weeks. In the survival arm of the study, mice were followed up until the age of 167 days.

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
