# Peer review of "Brain Protease Activated Receptor 1 Pathway: A Therapeutic Target in the Superoxide Dismutase 1 (SOD1) Mouse Model of Amyotrophic Lateral Sclerosis"

_ijms, 2020, doi:10.3390/ijms21103419_

Round 1
Reviewer 1 Report
In the article, authors explore the possibility of involving PAR1 receptors in the pathogenesis of amyotrophic
lateral sclerosis. A three-level study includes the inhibition of thrombin, PAR1 receptor blocking, and the Ras intracellular inhibition. The study is interesting and relevant with a verified protocol of experiments.
Few comments for transmission to the authors
1. PAR1 activation can induce different effects in cells. The heterotrimeric G proteins that interact with PAR1 have been identified, described and include Gαq11, Gαq, Gα12/13, etc. More over PAR1 can activated not only by thrombin, but also by other coagulation system-related proteases, such as factor Xa, activated protein C, plasmin, and kallikreins. All these will determinate the PAR1-induced intracellular pathway. At last, time many researchers point to the principle new mechanism of PAR1 activation “biased agonism”. Unfortunately, these aspects of PAR1-dependent regulation of cellular functions does not discuss in any way the article. Interestingly, more than 10 years ago the role of activated protein C in amyotrophic lateral sclerosis has investigated in SOD1-mutant mice (Zhihui Zhong, Hristelina Ilieva, Lee Hallagan, Robert Bell, Itender Singh, Nicole Paquette, Meenakshisundaram Thiyagarajan, Rashid Deane, Jose A Fernandez, Steven Lane, Anna B Zlokovic, Todd Liu, John H Griffin, Nienwen Chow, Francis J Castellino, Konstantin Stojanovic, Don W Cleveland, Berislav V Zlokovic. Activated Protein C Therapy Slows ALS-like Disease in Mice by Transcriptionally Inhibiting SOD1 in Motor Neurons and Microglia Cells. J Clin Invest. 2009 Nov;119(11):3437-49. doi: 10.1172/JCI38476. Epub 2009 Oct 19). Authors should critically analyze these data, discuss and compare them with their data. This information should be included in the introduction and discussion of article.
2. It is necessary to emphasize the prior role of PAR1 in the regulation of RAS compared with other receptors in astrocytes. It is not clear why the authors based on the data obtained using the RAS inhibitor concluded about PAR1 involvement in the pathogenesis ALS. For example, early it was shown EGFR activation led to reactive astrogliosis through activation of the Ras-Raf-ERK signaling pathway (Gao W-L at al., 2014).
Author Response
Reviewer 1
Comments and Suggestions for Authors
In the article, authors explore the possibility of involving PAR1 receptors in the pathogenesis of amyotrophic
lateral sclerosis. A three-level study includes the inhibition of thrombin, PAR1 receptor blocking, and the Ras intracellular inhibition. The study is interesting and relevant with a verified protocol of experiments.
Few comments for transmission to the authors
1. PAR1 activation can induce different effects in cells. The heterotrimeric G proteins that interact with PAR1 have been identified, described and include Gαq11, Gαq, Gα12/13, etc. More over PAR1 can activated not only by thrombin, but also by other coagulation system-related proteases, such as factor Xa, activated protein C, plasmin, and kallikreins. All these will determinate the PAR1-induced intracellular pathway. At last, time many researchers point to the principle new mechanism of PAR1 activation “biased agonism”. Unfortunately, these aspects of PAR1-dependent regulation of cellular functions does not discuss in any way the article. Interestingly, more than 10 years ago the role of activated protein C in amyotrophic lateral sclerosis has investigated in SOD1-mutant mice (Zhihui Zhong, Hristelina Ilieva, Lee Hallagan, Robert Bell, Itender Singh, Nicole Paquette, Meenakshisundaram Thiyagarajan, Rashid Deane, Jose A Fernandez, Steven Lane, Anna B Zlokovic, Todd Liu, John H Griffin, Nienwen Chow, Francis J Castellino, Konstantin Stojanovic, Don W Cleveland, Berislav V Zlokovic. Activated Protein C Therapy Slows ALS-like Disease in Mice by Transcriptionally Inhibiting SOD1 in Motor Neurons and Microglia Cells. J Clin Invest. 2009 Nov;119(11):3437-49. doi: 10.1172/JCI38476. Epub 2009 Oct 19). Authors should critically analyze these data, discuss and compare them with their data. This information should be included in the introduction and discussion of article.
We thank the reviewer for this important comment that strengthens the main study conclusion. The inverse association between thrombin and APC effects is well known, suggesting that either an increase APC, or thrombin inhibition may be of benefit in this disease. The role of APC in the PAR1 pathway and specifically in SOD1 mice has been added to the Introduction (page 2, lines 64-69) and Discussion (page 6, lines 196-201).
2. It is necessary to emphasize the prior role of PAR1 in the regulation of RAS compared with other receptors in astrocytes. It is not clear why the authors based on the data obtained using the RAS inhibitor concluded about PAR1 involvement in the pathogenesis ALS. For example, early it was shown EGFR activation led to reactive astrogliosis through activation of the Ras-Raf-ERK signaling pathway (Gao W-L at al., 2014).
We agree completely with the reviewer that the beneficial effect of the Ras inhibitor presently used in the study is not necessarily specific for the PAR1-thrombin pathway. However, in conjunction with the beneficial effects of inhibiting thrombin itself and PAR1 antagonism, these results indicate that this pathway is indeed important in the pathogenesis of this model. This has been stressed and discussed in the text (page 7, lines 264-268).
Reviewer 2 Report
This is an interesting manuscript and an extension of previous work published by this group on the potential role of the protease-activated receptor 1 (PAR1) pathway in the neurodegenerative disease amyotrophic lateral sclerosis (ALS). Here, the authors report on the effects of the pharmacological targeting of PAR 1 and related pathways.
Overall the article is very interesting and offers a contribution to ALS research, however, some concerns need to be addressed.
- My understanding is that the authors have used a SOD1G93A mouse carrying low-copy number they mention the provider Prof Yoram Groner, however, they should describe with more details this strain of mice, the mutant SOD1 copy number and the tissue characteristics.
I’m asking this because I don’t understand the GFAP staining, why is decreased in the mutant? It shouldn’t be the opposite? By looking at Fig.2 they have increased staining in the controls than the mutant. This is puzzling me, reactive gliosis is a feature of ALS, why in their model is not? What do they mean by “astrocytic morphological modification” (Fig. 2 legends)?
- Why they present data about the cerebellum and cortex? Cortical areas are affected in ALS, and they say that also the cerebellum is affected, which is probably true. Yet, I don’t understand why they don’t show any data of spinal cord or brainstem, the two most affected areas, at least in the SOD1G93A model. Why they show/talk about the cerebellum? What is their rationale?
- The treatment was able to alter thrombin activity and/or modify PAR1 expression? Why they look at thrombin and PAR1 in the whole brain and spinal cord, and then immunostaining in cortex and cerebellum?
Author Response
Reviewer 2 Comments and Suggestions for Authors
This is an interesting manuscript and an extension of previous work published by this group on the potential role of the protease-activated receptor 1 (PAR1) pathway in the neurodegenerative disease amyotrophic lateral sclerosis (ALS). Here, the authors report on the effects of the pharmacological targeting of PAR 1 and related pathways.
Overall the article is very interesting and offers a contribution to ALS research, however, some concerns need to be addressed.
- My understanding is that the authors have used a SOD1G93A mouse carrying low-copy number they mention the provider Prof Yoram Groner, however, they should describe with more details this strain of mice, the mutant SOD1 copy number and the tissue characteristics.
The strain used was based on a mouse from the Jackson Laboratories (number B6.Cg-Tg(SOD) with high number of copies of the mutated human SOD1 gene (Methods, page 9, lines 329-330). The transgene is expressed in many tissues of the brain, including neurons and glia (Gill 2019, page 7, lines 249-250). Since the animals’ phenotype is a major indication for mutant SOD1 copies we measured the lifespan, motor function deterioration and infertility. The detailed monitored motor function and infertility and survival data was found to be similar during all the duration of the experiment (670 days, Methods, page 9, lines 341-343).
I’m asking this because I don’t understand the GFAP staining, why is decreased in the mutant? It shouldn’t be the opposite? By looking at Fig.2 they have increased staining in the controls than the mutant. This is puzzling me, reactive gliosis is a feature of ALS, why in their model is not? What do they mean by “astrocytic morphological modification” (Fig. 2 legends)?
Recent studies demonstrated the cerebellum involvement in ALS pathology. Specifically, the cerebellum Purkinje cells degeneration was published in SOD1 mice (Afshar et al. 2017, page 7, lines 249-250). Although glial involvement was previously described in ALS pathology, to our knowledge there are no detailed descriptions of GFAP staining in the SOD1 mouse cerebellum. Our data is in line with some of the reports of astrocyte depletions in some areas in this model.
- Why they present data about the cerebellum and cortex? Cortical areas are affected in ALS, and they say that also the cerebellum is affected, which is probably true. Yet, I don’t understand why they don’t show any data of spinal cord or brainstem, the two most affected areas, at least in the SOD1G93A model. Why they show/talk about the cerebellum? What is their rationale?
Most of our focus in the study of PAR1 on astrocytes has been in areas such as the hippocampus, cortex and cerebellum which express the highest level of this receptor. We therefore focused on these areas in the present study and this is emphasized in the title of the paper.
- The treatment was able to alter thrombin activity and/or modify PAR1 expression? Why they look at thrombin and PAR1 in the whole brain and spinal cord, and then immunostaining in cortex and cerebellum?
As stated in the previous point, the focus of the study was in the areas of high PAR1 expression which are the cortex, hippocampus, and cerebellum. Since the spinal cord is heavily affected in the SOD1 mouse we also carried out some measurements in the spinal cord which displayed similar effects on PAR1. A detailed characterization of histology of PAR1 in the spinal cord is beyond the scope of the present manuscript.
Reviewer 3 Report
The study by Shavit-Stein et al., investigates a potential mechanistic pathway in amyotrophic lateral sclerosis (ALS). Using the SOD1 ALS mouse model, the authors suggest that a pathway between thrombin, PAR1, and downstream RAS signaling was potentially involved in the neuropathologic cascade found in ALS. While the data does show inhibiting this pathway at multiple levels ameliorates some of the pathologic phenotype, there are still several concerns with the manuscript.
- The authors begin by demonstrating that thrombin levels are higher in the SOD1 mouse model compared to controls. While it is suggested that this is through blood brain barrier disruption, there is no data to support this claim. Simple stains with thrombin, IgG, or albumin antibodies should be able to demonstrate that there is BBB disruption and thrombin can even interact with brain resident proteins.
- Additionally, despite claiming thrombin and PAR1 exist in a pathway, no connection is made. It would be helpful to plot thrombin levels against PAR1 levels and show that there is a correlation between the two.
- In line with point 2, the specific relationship between PAR1 and thrombin is unclear. It seems like there exist a pathway where thrombin activates PAR1, however, there are lower levels of PAR1 in SOD1 mice, which the thrombin levels are increased. Can this be clarified a bit more?
- The staining present in Figure 2 is not convincing that PAR1 colocalizes with astrocytes. Higher power images are probably needed for this. Additionally, the authors should refrain from making statements found in line 113-117 where they claim the staining intensities or amounts are different, but don’t have any sort of analysis to back up the claims. Quantitations are needed for this.
- Could the loss of PAR1 just simply reflect the loss of neurons or astrocytes during disease? Standardizing the PAR1 levels to the number of neurons and/or astrocytes would help determine if there is actually a reduction in PAR1 levels in each cell or if it based on overall cell loss.
- The graphs in figure 3 are hard to interpret on their own. It would be helpful to include stars denoting significance as opposed to just referencing them in the text.
Author Response
Reviewer 3 Comments and Suggestions for Authors
The study by Shavit-Stein et al., investigates a potential mechanistic pathway in amyotrophic lateral sclerosis (ALS). Using the SOD1 ALS mouse model, the authors suggest that a pathway between thrombin, PAR1, and downstream RAS signaling was potentially involved in the neuropathologic cascade found in ALS. While the data does show inhibiting this pathway at multiple levels ameliorates some of the pathologic phenotype, there are still several concerns with the manuscript.
- The authors begin by demonstrating that thrombin levels are higher in the SOD1 mouse model compared to controls. While it is suggested that this is through blood brain barrier disruption, there is no data to support this claim. Simple stains with thrombin, IgG, or albumin antibodies should be able to demonstrate that there is BBB disruption and thrombin can even interact with brain resident proteins.
As indicated by the reviewer, the increased brain thrombin levels may be due to either intrinsic local production or to leakage from the blood through a disrupted BBB. This was well described in several previous publications (Bushi 2015, Shavit-Stein 2015, Itsekson-Hayosh 2016). Although there are reports of Blood spinal cord barrier disruption (Andjus 2009, Garbusova-Davis 2014) it is not clear that there is a significant BBB disruption in this ALS model. Although the source of excess thrombin in the SOD1 model is not yet clear, much of it may be from intrinsic thrombin production by astrocytes. This was added to the text (Discussion, page 6, lines 202-206).
Studying the interaction of thrombin with brain resident proteins is very interesting but this type of study is not simple, since antibodies against thrombin cross react with prothrombin which is not active enzymatically and may be a confounder.
- Additionally, despite claiming thrombin and PAR1 exist in a pathway, no connection is made. It would be helpful to plot thrombin levels against PAR1 levels and show that there is a correlation between the two.
Indeed, a correlation between thrombin levels and PAR1 may support the causality. However, in the present experimental design it was not possible to perform these measures in the same animals since the thrombin activity measurements were conducted on brain slices derived from other animals from the ones used for whole brain western blot homogenates.
- In line with point 2, the specific relationship between PAR1 and thrombin is unclear. It seems like there exist a pathway where thrombin activates PAR1, however, there are lower levels of PAR1 in SOD1 mice, which the thrombin levels are increased. Can this be clarified a bit more?
We agree with the reviewer that this point is complex and requires clarification: High levels of thrombin cause over activation of PAR1 , receptor consumption and internalization and probably astrocyte dysfunction and death. This correlation between high thrombin levels and low PAR1 levels was observed and published before (Itsekson 2016) and is supported by pharmacological studies (Coughlin 1999). The relevant references were added to the manuscript (Discussion, page 6, lines 207-211).
- The staining present in Figure 2 is not convincing that PAR1 colocalizes with astrocytes. Higher power images are probably needed for this. Additionally, the authors should refrain from making statements found in line 113-117 where they claim the staining intensities or amounts are different, but don’t have any sort of analysis to back up the claims. Quantitations are needed for this.
Indeed, the reviewer is correct in these comments. This observation was qualitative (based on 3 to 4 animals per group) rather than quantitative. This was added to the Methods (page 10, line 401) and relevant figure legend (page 4, line 129).
- Could the loss of PAR1 just simply reflect the loss of neurons or astrocytes during disease? Standardizing the PAR1 levels to the number of neurons and/or astrocytes would help determine if there is actually a reduction in PAR1 levels in each cell or if it based on overall cell loss.
Indeed, the levels of PAR1 are dependent on several factors including the number of astrocytes, the number of astrocyte processes, activation of astrocytes and thrombin levels. Simple quantification of astrocyte numbers would correlate with PAR1 loss but would not unravel the role of excess thrombin in this. We therefor modulated the thrombin – PAR1 pathway at 3 different points and the evidence from this indicates that this pathway plays a significant role in disease progression.
- The graphs in figure 3 are hard to interpret on their own. It would be helpful to include stars denoting significance as opposed to just referencing them in the text.
We have addressed this issue. Symbols indicating significance were added to Figure 3.
Round 2
Reviewer 2 Report
accept
Author Response
Thanks
Reviewer 3 Report
Although the authors have clarified some of the points regarding the interactions between PAR1 and thrombin, they elected not to perform any of the recommended additional experiments. There are still two major issues with this manuscript in which the data, and subsequent interpretations, are dependent on and need to be addressed.
- The images presented in figure 2 need to be improved. The PAR1/GFAP colocalization is not obvious from the images presented. Either better representative images or high power confocal orthogonal images are needed to demonstrate overlap. The argument that “qualitative analysis” shows change in staining means nothing if the images do not accurately portray this. Additionally, the variability in staining intensities should be more consistent between images.
- All the findings in this paper could be explained by loss of neurons or astrocytes rather than decrease in PAR1 levels. It is imperative that the authors take this into account. It should simple to count the number of astrocytes and neurons to standardize measurements.
Author Response
Although the authors have clarified some of the points regarding the interactions between PAR1 and thrombin, they elected not to perform any of the recommended additional experiments. There are still two major issues with this manuscript in which the data, and subsequent interpretations, are dependent on and need to be addressed.
- The images presented in figure 2 need to be improved. The PAR1/GFAP colocalization is not obvious from the images presented. Either better representative images or high power confocal orthogonal images are needed to demonstrate overlap. The argument that “qualitative analysis” shows change in staining means nothing if the images do not accurately portray this. Additionally, the variability in staining intensities should be more consistent between images.
Indeed, high power confocal images better demonstrate the PAR1/GFAP colocalization. Figure 2 was edited accordingly (pages 4-5 lines 140-143).
- All the findings in this paper could be explained by loss of neurons or astrocytes rather than decrease in PAR1 levels. It is imperative that the authors take this into account. It should simple to count the number of astrocytes and neurons to standardize measurements.
We agree with the reviewer, this issue is very important. We were able to address this issue by 2 approaches: a semi-quantitative analysis for Purkinje cells density based on PAR1 surrounding staining. This indeed indicates a small non- significant reduction in cell bodies count (this was added to the text: Results: pages 3-4, lines 115-135, Methods, page 11, lines 418-420).
The glia issue is more complex and after carefully re-assessing our results we were able to discern 2 different phenomena:
- In the Granular layer we indeed detected a change in astrocytic morphology that may corresponds to the known gliosis in the SOD1 model. However, near the Purkinje cell-bodies we detected a decreased staining of the astrocyte tips that ensheath the neuron cell body.
- In the Molecular layer we detected a significant loss of radial astrocytic fibers probably corresponding to the Bergman glial fibers. This may indicate a change of structure or function of these cells.
The relevant loss of PAR1 surrounding Purkinje cell bodies does not seem to be directly linked to neuronal loss or to general astrocytic loss. It may be related to regional changes in specific astrocyte sub-types such as the Bergmann glia but further characterization of these specific astrocyte changes is beyond the scope of the present study. We have added these points to the results (pages 3-4 lines 115-135) and the discussion (page 8, lines 245-248).
Round 3
Reviewer 3 Report
The authors have sufficiently addressed my questions.